# Rapid 3D phenotyping of chick embryo liver development at HH22-HH41 embryonic stages using X-ray microcomputed tomography with PTA staining

Igor Rzhepakovsky[1], Sergei Piskov[1], Svetlana Avanesyan[1], Marina Sizonenko[1], Lyudmila Timchenko[1], Magomed Shakhbanov[1], Gloria Nassali[2], Idrisa Kiryowa[3], Andrey Nagdalian[1]*

1 North-Caucasus Federal University, Stavropol, Russia, 2 Makerere University, Kampala, Uganda, 3 Institute for Precise Mechanics and Optics (ITMO), St. Petersburg, Russia

* anagdalian@ncfu.ru

## Abstract

Animal models, particularly the chicken embryo (CE), remain crucial for advancing developmental biology and medicine. Liver embryogenesis, a complex and tightly regulated process, is particularly susceptible to developmental abnormalities. This study presents a comprehensive 2D and 3D µCT analysis of CE liver development (HH22–HH41) using a refined 1% phosphotungstic acid (PTA) staining protocol. Our methodology yielded high-resolution visualization of liver microstructures, including intricate vascular networks, with image quality and contrast comparable to histological analysis. Quantitative assessment revealed a critical period of rapid liver growth between incubation days 6 and 9 (HH29–HH35), followed by the stabilization of hepatic vascular volume by day 10 (HH36). The ease of structural orientation in µCT datasets, enhanced by 3D renderings, further underscores the technique's utility. By establishing µCT as an effective tool for liver imaging, this study provides essential normative data on the liver's spatial organization, morphology, and vascular architecture during embryogenesis, thereby opening new avenues for research in embryology, teratology, pharmacology, and toxicology.

## 1. Introduction

Advances in developmental biology and medicine are intrinsically linked to the utilization of animal models for the formulation and validation of experimental hypotheses [1]. While organoid systems offer promising avenues for modeling organogenesis, significant technological hurdles persist, necessitating the continued reliance on animal models for in-depth investigations into organ embryogenesis [2,3]. The chicken embryo (CE) has emerged as a particularly valuable model in this domain [4,5], owing to its notable developmental similarities to mammalian embryos and the

**Data availability statement:** All relevant data are within the paper.

**Funding:** The author(s) received no specific funding for this work.

**Competing interests:** The authors have declared that no competing interests exist.

accessibility it affords for comprehensive research across all stages of embryonic development [6]. Furthermore, the *in ovo* development of the CE, subject to environmental influences, renders it an ideal system for dissecting the impact of various factors on organ embryogenesis [7].

Among developing organs, the liver garners significant scientific attention. As one of the earliest and most structurally intricate organs to develop in the CE [8], the embryonic liver exhibits immunocompetence and a marked sensitivity to environmental perturbations. Given its pivotal roles in numerous critical functions, alterations in its size, morphology, or function during development frequently result in embryonic lethality [9]. Moreover, developmental anomalies of the liver are implicated in various congenital and perinatal hepatic diseases, underscoring the importance of studying its morphogenesis to elucidate both its formation and functional capacity. Notably, the liver occupies a substantial portion of the abdominal cavity throughout much of the prenatal period, establishing close spatial relationships with and influencing the development of adjacent anatomical structures [10,11].

Despite certain unique features in the formation of its hepatic sinusoids, tubules, and cords, the liver of the CE exhibits a closer anatomical resemblance to the human embryonic liver compared to non-mammalian models. Consequently, the CE continues to be a subject of considerable scientific interest and a significant contributor to our understanding of developing liver morphology [12]. Particular emphasis is placed on the embryonic and pre-fetal periods of CE development, specifically the Hamburger-Hamilton (HH) stages 22–41. During this critical window, the liver undergoes a complex morphogenetic program involving the coordinated execution of numerous events, where even subtle deviations can precipitate developmental abnormalities. The HH22-HH41 stages are characterized by the maximal increase in liver mass and the initiation and peak of hepatic hematopoiesis. By the HH41 stage, the liver's histostructure closely approximates that of an adult hen [13]. While many developmental changes occur at the cellular level, significant anatomical transformations are also evident at the macroscopic scale [9].

The transient nature of liver morphological changes, coupled with the intricate spatial organization of the organ, presents a considerable challenge for visualization. Furthermore, the inherent limitations of two-dimensional (2D) imaging modalities impede a comprehensive understanding of the complex structural and volumetric dynamics of organogenesis. Transitioning to three-dimensional (3D) models offers the potential to observe liver development in a dynamic manner and to perform quantitative analyses of morphogenesis through model reconstruction [14].

The integration of individual model organisms with advanced 3D imaging techniques has already yielded novel insights into morphogenetic processes in the developing liver [15,16]. However, the molecular mechanisms governing these processes are only beginning to be elucidated in detail, and the precise interactions and connectivity between hepatocyte strands, bile ducts, and vessels during liver development remain largely unexplored. A comprehensive understanding of the 3D tissue architecture of the developing CE liver is a fundamental prerequisite for deciphering the

pathophysiology of developmental abnormalities and for the *in vitro* regeneration of liver tissue for regenerative medicine. Nevertheless, this aspect remains insufficiently investigated and developed.

The current paucity of accurate measurement data for individual CE liver structures necessitates the application of high-quality imaging modalities. Researchers are actively employing and refining various technical approaches to acquire high-resolution 3D images of the developing liver [17,18]. X-ray microtomography (µCT) is of particular interest in this context. This non-destructive imaging technique offers high speed, high resolution, quantitative assessment of tissue size and morphology, and rapid image acquisition [19,20]. Surprisingly, the potential of µCT for imaging analysis of CE liver morphogenesis has not been directly evaluated. This study aims to address the following critical questions:

1. Can µCT effectively visualize the morphological evolution of the CE liver during the embryonic and prenatal periods?

2. Is µCT a viable technique for the volumetric quantitative assessment of hepatogenesis in the CE?

3. Does 3D visualization of the liver provide novel and valuable insights for the CE model in developmental biology?

This investigation focuses on providing novel insights into the morphogenesis of the CE liver and proposes the development of methodological principles, alongside the creation of a comprehensive set of spatial and quantitative µCT data for assessing liver morphogenesis across the embryonic stages HH22-HH41. This will generate a digital representation of the precise shapes and topography of liver structures, enabling the identification of subtle changes in CE liver morphology. Thus, the primary objective of this work was the visualization and quantitative analysis of CE liver morphogenesis at the HH22–HH41 stages of embryogenesis utilizing µCT.

## 2. Materials and methods

### 2.1 Chemicals

For the experiment, formalin solution, neutral buffered 10%, isopropyl alcohol ≥99.7%, ethyl alcohol 95%, medical paraffin Histomix, hematoxylin and eosin were obtained from Biovitrum (Russia) and phosphotungstic acid hydrate 99.99% was purchased from Sigma-Aldrich (USA).

### 2.2 Embryo preparation

The study was conducted in accordance with the Helsinki Declaration. The design of the experiments was approved by the local Ethics committee of North Caucasus Federal University (Protocol No. 003 dated 03 August 2023). The study was conducted within the framework of the strategic academic leadership program "Priority-2030" using the equipment of the Center for Collective Use of the North Caucasus Federal University.

Fertilized Hysex Brown eggs, each weighing between 50 and 55 grams, were procured from the commercial hatchery of Agrokormservice Plus (Giaginskaya village, Republic of Adygea, Russia). The eggs were incubated under controlled conditions (T = 37.5 °C, W = 50%) in a digital incubator Rcom Maru Deluxe Max 380 (Autoelex CO in Gyeongsangnam-do, Korea). The trays were automatically rotated every two hours. Each egg was inspected daily to confirm CE viability using the PKYA-10 ovoscope (Premier, Moscow, Russia). The stage of development was determined using the Hamburger and Hamilton (HH) system [21]. CE selection was carried out in the amount of 5 samples for each day of development from the 4th (HH 22–24) to the 15th (HH41) days.

According to the guidelines for euthanasia provided by Bjørnstad [22] and Underwood and Anthony [23], CE were euthanized using two methods depending on embryonic stages. For stages from HH22–24 to HH33–34, CE were cooled at 4 °C for 4 hours. For stages from HH36 to HH41, CE were exposed to 70% $CO_2$ for 30 minutes. To extract the CE, a section of the eggshell and the inner membrane in the area of the egg's blunt pole above the air chamber were removed. The edge of this area was previously marked during ovoscopy. After extraction, CE were washed with a saline solution and fixed in neutral buffered formalin for 72 hours. CE with structural abnormalities or mechanical injuries were excluded from the study.

Macroscopic examination of CE was performed using Axio Zoom light microscope (Carl Zeiss Microscopy, Germany). The macroscopic images were obtained by AxioCam MRc5 camera and processed with Zen 2 Pro software.

## 2.3 Stain

CE were stained with phosphotungstic acid (1% PTA) as contrast agent. CE from days 4–8 (HH22–HH34), fixed in a 10% buffered formalin solution for 72 hours, were washed under running water for 12 hours. CE from days 9–15 (HH35–HH41), fixed in a 10% buffered formalin solution for 96 hours, were washed under running water for 24 hours. All CE were dehydrated in replaceable portions of ethanol: 30% for 2 hours, 50% for 2 hours, and 70% for 12 hours. Afterwards, the dehydrated CE were placed in a 1% PTA solution at a ratio of 1:20 and kept at 40 °C for 24 hours (HH22–HH34) or 96 hours (HH35–HH41) as was shown in the previous work [24].

## 2.4 µCT scanning

For µCT scanning, different types of tubes were utilized depending on the developmental stage of the samples. Specifically, Eppendorf Safe-Lock Tubes (2 mL, colorless, polypropylene) were used for 4–7-days CE (HH22-HH32). Servicebio Centrifuge Tubes (15 mL, colorless, polypropylene) were applied for 8–12-days CE (HH33-HH38). Finally, Servicebio Centrifuge Tubes BioBased (50 mL, colorless, polypropylene) were used for 13–15-days CE (HH39-HH41). The test tubes containing CE in 70% ethanol solution were then transferred to the Skyscan 1176 microtomograph (Bruker, Kontich, Belgium). The samples were securely fixed in place using foam retainers. It is worth noting that the use of a 70% ethanol solution is justified by its low radiopacity. This allowed clear visualization of the lowest contrast parts of the CE [25].

For the cone-beam computed tomography (CBCT) scan, we used an 11-megapixel camera (4000×2672 pixels) and rotated it by 180 degrees, taking three images at each step (0.3 degrees per step). This allowed us to achieve an isometric spatial resolution of 8.87 µm for the period from the 4th to the 12th day (HH22-HH38) and 17.74 µm for the period from the 13th to the 15th day (HH39-HH41).

CE scanning was performed rotating an 11-megapixel camera (4000×2672 pixels) by 180° (0.3°/step) with averaging of three images per step, resulting in an isometric spatial resolution 8.87 µm (for 4–12 day, HH22-HH38) and 17.74 µm (for 13–15 day, HH39-HH41). The radiation penetration level for most planes in CE scanning was 30–50%. During image reconstruction, the grey value histogram of all images was adjusted to include the contrast region, defined by the minimum and maximum grey values. This approach enhanced both the overall and differentiated contrast of the CE parts. Applying a wider filter increased the overall contrast level of the CE. This helped to clearly distinguish the CE from the surrounding space [24].

All manipulations of the CE µCT scans were performed utilizing software provided by Bruker (Kontich, Belgium). Raw image stacks were processed and reconstructed into 3D datasets using NRecon software (version 1.7.4.2). This reconstruction process requires approximately two hours per sample tube. Subsequent analysis steps, including postprocessing, alignment, spatial orientation (x, y, z), mapping of X-ray contrast profiles, and the delineation of specific regions of interest within the reconstructed volumes, were conducted using DataViewer software (version 1.5.6.2). Three-dimensional image visualization was achieved using CTvox software (version 3.3.0r1403). Morphometric analyses and the assessment of X-ray density within various CE structures were performed using CT-analyser software (version 1.18.4.0), with analysis protocols informed by the methodology described by Tahara and Larsson [26].

Volume segmentation of the 3D image datasets was performed using an algorithm provided by Bruker-microCT (Kontich, Belgium). This ensured the consistent identification and precise allocation of the areas of interest. Segmented structures were saved as individual volume files. For the majority of quantitative analyses presented herein, the organ or specific region of interest was isolated through the creation of a negative space. This involved setting the pixel values within the target region to zero contrast, effectively removing it from the surrounding tissue, and propagating this void throughout the Z-plane. This method allowed for the selective capture and independent quantification of tissues or fluid

volume within a specific organ. Multiple labeled volumes were derived from each CE and quantified using the same methodology. For each region or tissue analyzed, data from a minimum of five individual CEs were included to ensure statistical robustness.

## 2.5  Histological preparation

To ensure accurate anatomical identification and interpretation within the μCT datasets, particularly concerning the structures of the CE liver, we employed a comparative approach with corresponding histological sections, following established methodologies [26–30]. This involved direct visual comparison of the μCT images with stained histological sections of embryos at corresponding developmental stages.

For histological processing, CEs were collected at the designated developmental time points, immediately fixed, and subsequently washed. Following these initial steps, the CE samples underwent a graded dehydration series using isopropyl alcohol and were then embedded in Histomix medical paraffin (Biovitrum, St. Petersburg, Russia). Serial histological sections with a thickness of 6 μm were prepared using a rotary microtome NM 325 (Thermo Fisher Scientific, Waltham, US). The prepared sections were subjected to hematoxylin and eosin (H&E) staining according to standard protocols [31]. Briefly, the paraffin sections were dewaxed through two cycles of incubation in xylene, followed by rehydration through a descending ethanol series (95%, 80%, and 70%) and a final wash in distilled water. Subsequently, the sections were incubated in a hematoxylin solution for 3 minutes, rinsed thoroughly with tap water, and then incubated in a 1% aqueous eosin solution for 5 minutes. After a further rinse with distilled water to remove excess stain, any residual water was carefully removed. The sections were then dehydrated through 96% ethanol and xylene before being permanently mounted using Vitrogel mounting medium (Biovitrum, St. Petersburg, Russia). Photomicrographs of the prepared slides were captured using Axio Zoom V16 and Axio Imager 2 (A2) research-grade microscopes (Carl Zeiss Microscopy, Oberkochen, Germany). Image acquisition was performed using a specialized AxioCam MRc5 camera and Zen 2 Pro software.

## 2.6  Statistical data processing

For each embryonic stage, five CE were used. To visualize 2D and 3D structures, as well as radiopacity profiles, the most representative materials were selected. The selection process involved excluding digital materials with mechanical or digital defects that could have resulted from staining or scanning. Segmentation and quantification were carried out according to the method of Kim et al. [32]. Skyscan 1176 (software platform Bruker, Kontich, Belgium) running on a Windows 7 Professional (Microsoft Corp., Redmond, WA, USA) workstation with 32 Gb of RAM and an Nvidia Quadro K 4000 graphics card (Nvidia Corp., Santa Clara, CA, USA) was used for μCT data processing. The assessment of individual differences in the samples was carried out using statistical analysis using ANOVA, followed by Tukey post hoc testing using $p < 0.05$ as a significance threshold.

## 3.  Results and discussion

Phosphotungstic acid (PTA) enhancement of X-ray contrast of embryonic tissues in μCT imaging is a well-established technique, offering an alternative to contrast agents containing iodine, osmium tetroxide, and other heavy metals [33]. However, this method presents certain limitations, notably the relatively low penetration efficiency of PTA, and the potential for tissue deformation at higher concentrations [34]. Consequently, ongoing efforts are directed towards refining PTA-based contrast methods for diverse biological specimens, including organs and tissues [16,35–38].

The efficacy of any contrast agent hinges on its ability to generate a discernible differential signal [39]. Our previously validated 1% PTA contrast protocols for whole CE and extraembryonic tissues enabled the achievement of a pronounced differential contrast for the majority of soft tissue structures, yielding a high level of detail comparable to that observed in histological preparations. Concurrently, we confirmed the substantial general and differential radiopacity of the liver, establishing the necessary foundation for the investigation of its microstructures [24,25]. Several inherent advantages position

μCT as a valuable complement to traditional histological analysis. Histology necessitates the destructive sectioning of a three-dimensional biological sample, and the intricate processes of probe preparation and sectioning can introduce artifacts such as tears, fractures, or tissue folds [40]. Furthermore, the reconstruction of an approximation of the original 3D structure from a series of 2D histological images demands a complex process of image acquisition and alignment. Despite recent advancements in automated 3D registration and alignment of serial sections, this process still involves the destruction of the original sample and can only minimize, but not entirely eliminate, alignment errors and artificial deformations of biological structures [41].

The non-destructive investigation of the embryonic liver in various laboratory animals represents a critical pursuit in modern science. Such studies often focus on accurately determining the organ's location, overall and regional dimensions, lobation patterns, and the quantification of vessel tortuosity and density. Notably, the inner walls of blood vessels are clearly discernible when the lumen is devoid of blood, facilitating the tracing of vessel branching patterns. Even when the lumens are filled with blood, inherent contrast differences allow for enhanced visualization, potentially enabling the creation of angiographic representations [16,42–45]. The permeability and relatively small size of the embryos collected across various developmental stages permit the application of rapid processing and visualization methodologies. This approach proves valuable for observing gross morphological changes and for constructing comprehensive atlases and digital archives using μCT [24,25,46]. Accordingly, the calculated radiopacity and visualized volumes of the liver and its vascular system during CE development from days 4–15 (HH22-HH41) are presented in Table 1 and Fig 1.

Our results indicate a decrease in the total X-ray density of the CE during development from days 4–8 (HH22-HH34). This observation is likely associated with the rapid growth of the embryo and a potential decrease in PTA permeability. However, the 234.5% decrease in X-ray density was not directly correlated with the over 35-fold increase in recorded volume. A significant increase in the X-ray density of the CE on day 9 (HH35) was observed, which is attributed to a modification in the staining protocol (96 hours instead of 24 hours). Subsequent changes in overall contrast were influenced not only by growth but also by the formation of various organs and tissues, particularly the skeletal system, and remained at a high level throughout the studied developmental stages. In contrast to the whole embryo, the radiopacity of the liver was more pronounced. For instance, liver radiopacity exceeded 12,000 HU on days 5 and 6 (HH25-HH29), potentially due to an increase in the relative volume of liver parenchyma compared to day 4 (HH22-HH24) [47]. On days 7 and 8 (HH30-HH34), the X-ray density decreased by almost twofold, a finding corroborated by a significant increase in the total volume of the liver. On days 9–10 (HH35-HH36), a repeated increase in density above 12,000 HU was observed, again associated with the change in the staining protocol. Further observations did not reveal significant changes in liver X-ray density, which remained within the range of 8104–9083 HU, indicating the effectiveness of the developed protocol.

Analysis of the changes in the visualized volume of the liver during development revealed two notable periods of rapid growth: a 695% increase from the 4th to the 5th day and a 385% increase from the 6th to the 7th day. Another interesting finding was the stabilization of growth in both the CE and the liver from the 10th to the 11th day, consistent with previous studies [21,24,25].

Most importantly, the analysis of the ratio between CE and liver growth rates during the studied developmental periods is particularly important. Fig 1(A) demonstrates that from the 4th to the 6th day of incubation (HH22-HH29), the growth rates of the CE and liver are similar, and the ratio of the visualized organ volume within the CE does not vary significantly (p > 0.05). However, from the 6th to the 9th day of incubation (HH29-HH35), a more intensive liver growth occurs, increasing from 0.58 ± 0.03% to 1.91 ± 0.1% (p < 0.05). From the 9th to the 15th day of incubation (HH35-HH41), the ratio of CE and liver growth rate does not change significantly (p > 0.05). This observation aligns with some studies suggesting a stabilization of the micromorphological characteristics of the liver during this period. For instance, Wong and Cavey [13] observed that between HH36 and HH40 in CE development, the volume of liver cells increases, and hepatocytes attain a relatively uniform size.

**Table 1. X.ray density and visualized volume of the liver of CE (HH22-HH41 embryonic stages), n = 5, M ± SD.**

| Embryonic stages | Chick embryo | | Liver | | | | |
|---|---|---|---|---|---|---|---|
| | X-ray density, HU | visualized volume, mm³ | x-ray density of the organ, HU | visualized volume of the organ, mm³ | visualized volume of the organ in CE, % | visualized volume of the vessels, mm³ | visualized volume of the vessels in liver, % |
| 4th day, HH22-HH24 | 6610.0 ± 311 | 15.8 ± 1.0 | 9770.0 ± 479 | 0.082 ± 0.006 | 0.51 ± 0.03 | 0.037 ± 0.002 | 45.1 ± 2.3 |
| 5th day, HH25-HH27 | 5256.0 ± 316 | 105.2 ± 4.0 | 12337.0 ± 648 | 0.57 ± 0.03 | 0.54 ± 0.03 | 0.14 ± 0.006 | 24.6 ± 1.2 |
| 6th day, HH28-HH29 | 5126.0 ± 325 | 146.4 ± 9.0 | 12887.0 ± 636 | 0.85 ± 0.05 | 0.58 ± 0.03 | 0.22 ± 0.01 | 25.9 ± 1.3 |
| 7th day, HH30-HH32 | 2444.0 ± 104 | 360.8 ± 21.0 | 6805.0 ± 312 | 3.28 ± 0.11 | 0.9 ± 0.05 | 0.95 ± 0.05 | 29.0 ± 1.5 |
| 8th day, HH33-HH34 | 2819.0 ± 160 | 565.3 ± 36.0 | 6758.0 ± 293 | 6.5 ± 0.21 | 1.15 ± 0.06 | 1.81 ± 0.09 | 27.9 ± 1.4 |
| 9th day, HH35 | 5218.0 ± 312 | 747.0 ± 45.0 | 12907.0 ± 624 | 14.3 ± 0.5 | 1.91 ± 0.1 | 2.66 ± 0.1 | 18.7 ± 0.94 |
| 10th day, HH36 | 4883.0 ± 298 | 1157.8 ± 73.0 | 12626.0 ± 678 | 23.9 ± 1.4 | 2.06 ± 0.11 | 3.87 ± 0.2 | 16.2 ± 0.85 |
| 11th day, HH37 | 3476.0 ± 232 | 1037.7 ± 89.0 | 8888.0 ± 487 | 21.8 ± 1.3 | 2.1 ± 0.12 | 3.73 ± 0.21 | 17.1 ± 0.9 |
| 12th day, HH38 | 3225.0 ± 235 | 2180.0 ± 164.0 | 8358.0 ± 444 | 41.2 ± 2.0 | 1.89 ± 0.1 | 5.86 ± 0.33 | 14.2 ± 0.72 |
| 13th day, HH39 | 3046.0 ± 279 | 2847.6 ± 175.0 | 9083.0 ± 491 | 58.7 ± 4.1 | 2.06 ± 0.11 | 8.3 ± 0.35 | 14.1 ± 0.71 |
| 14th day, HH40 | 3614.0 ± 325 | 4585.9 ± 234.0 | 9336.0 ± 470 | 86.8 ± 5.0 | 1.89 ± 0.1 | 11.8 ± 0.42 | 13.6 ± 0.68 |
| 15th day, HH41 | 4175.0 ± 341 | 6052.1 ± 295.0 | 8104.0 ± 499 | 102.1 ± 5.2 | 1.69 ± 0.09 | 12.6 ± 0.51 | 12.3 ± 0.62 |

M is mean; SD is standard deviation.

We also quantified the visualized volume of the liver's circulatory system during the studied periods of CE development and calculated its ratio to the total visualized liver volume. As shown in Fig 1(B), a significant decrease in the volume occupied by the circulatory system was recorded from the 4th to the 5th day of incubation (HH22-HH27), from 45.1 ± 2.3% to 24.6 ± 1.2% ($p < 0.05$). This finding is supported by morphological observations indicating a significant increase in liver parenchyma during this period. From day 5 to day 6 (HH27-HH29), the volume of the liver occupied by the circulatory system remained relatively stable ($p > 0.05$), increasing by day 7 (HH30-HH32) to 29.0 ± 1.5% ($p < 0.05$) and showing no significant change until day 8 of incubation (HH33-HH34) ($p > 0.05$). A significant decrease in the volume occupied by the circulatory system was again recorded from the 8th to the 9th days of incubation (HH34-HH35), from 27.9 ± 1.4% to 18.7 ± 0.94% ($p < 0.05$). Subsequently, from the 9th to the 15th days of incubation (HH35-HH41), the volume occupied by the circulatory system gradually decreased from 18.7 ± 0.94% to 12.3 ± 0.62% ($p < 0.05$), with minor, statistically insignificant variations between individual days ($p > 0.05$). These results suggest a gradual increase in the volume and transformation of liver parenchyma during development [13,21].

The subsequent phase of this study involved the visualization and identification of visceral organs within the thoracic cavity, as well as the macro- and microstructures of the liver during μCT examination of CEs (days 4–15, HH22-HH41), in direct comparison with corresponding histological sections (Fig 2). Furthermore, Figs 3–14 present representative microtomographic images of the thoracic cavity organs, cross-sectional images of the liver, and isosurface 3D renderings of the liver and its vasculature, all of which were verified and confirmed through parallel histological analysis. All major organs of the abdominal cavity of CEs from the 4th to the 15th days (HH22-HH41) were clearly visualized and corresponded to the normative indicators for the respective stages of embryogenesis [21].

The liver is situated within the thoracic cavity, with the heart positioned between its lobes, extending from right to left. The left and right lungs are located dorsally and superiorly on either side. Paired mesonephroses are also visualized dorsally but at a more caudal level. The stomach is adjacent to the left lobe of the liver, with the small intestine located inferiorly. Beginning on the 7th day of incubation, the spleen becomes distinctly visible on the left side, situated between the liver, stomach, and mesonephros.

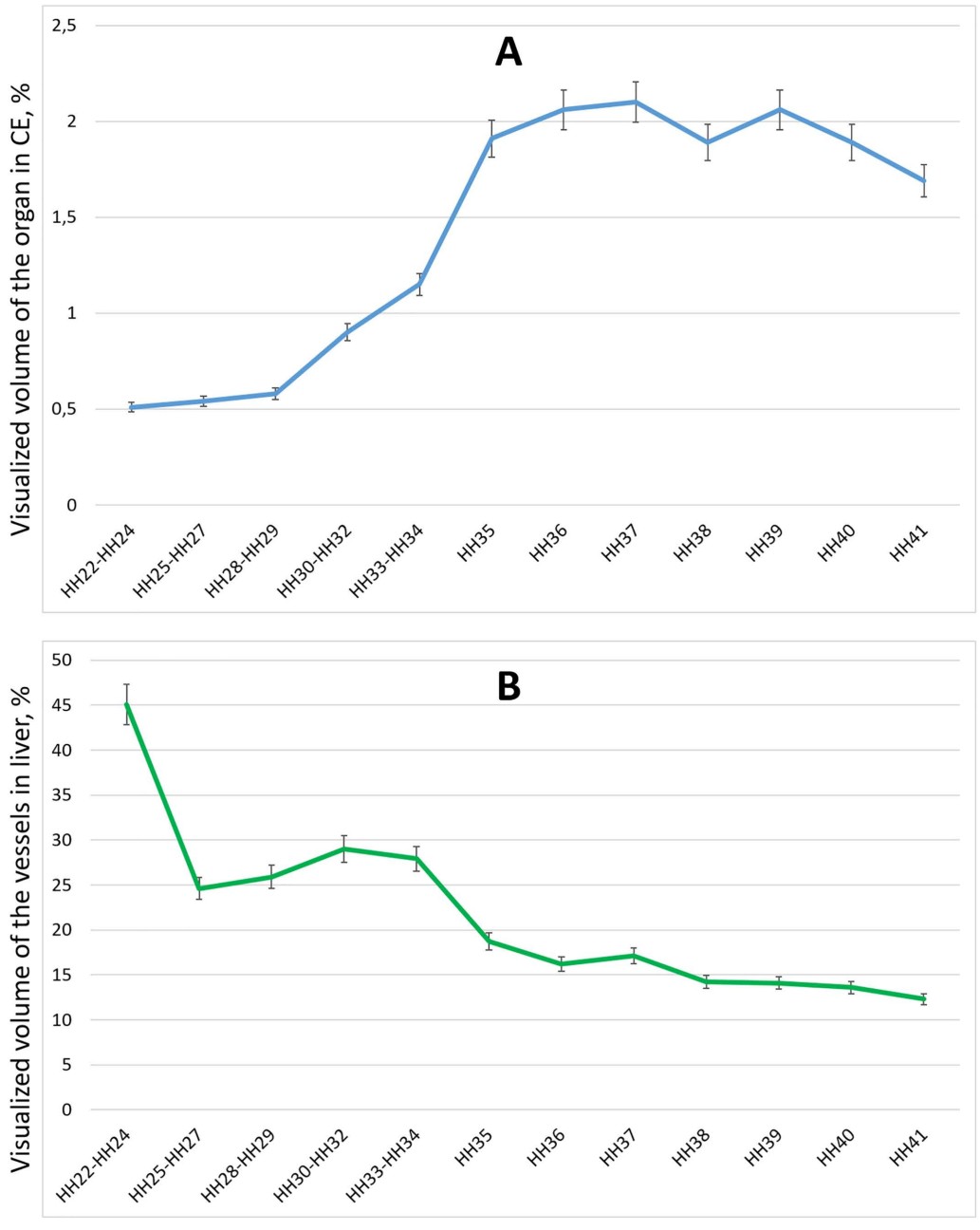

**Fig 1. Dynamics of visualized volume of the organ in CE (A) and visualized volume of the vessels in liver (B) at HH22-HH41embryonic stages, n = 5, M ± SD.**

On the 4th day of incubation (HH22-HH24), analysis of representative cross-sectional liver images and isosurface 3D renderings of the liver and its vessels allowed for the identification and clear visualization of the left lobe, right lobe, umbilical vein, ductus venosus, and inferior vena cava in various projections. At this stage, the right lobe constitutes a significant portion of the liver and encompasses the incoming umbilical vein and the outgoing inferior vena cava. The ductus venosus

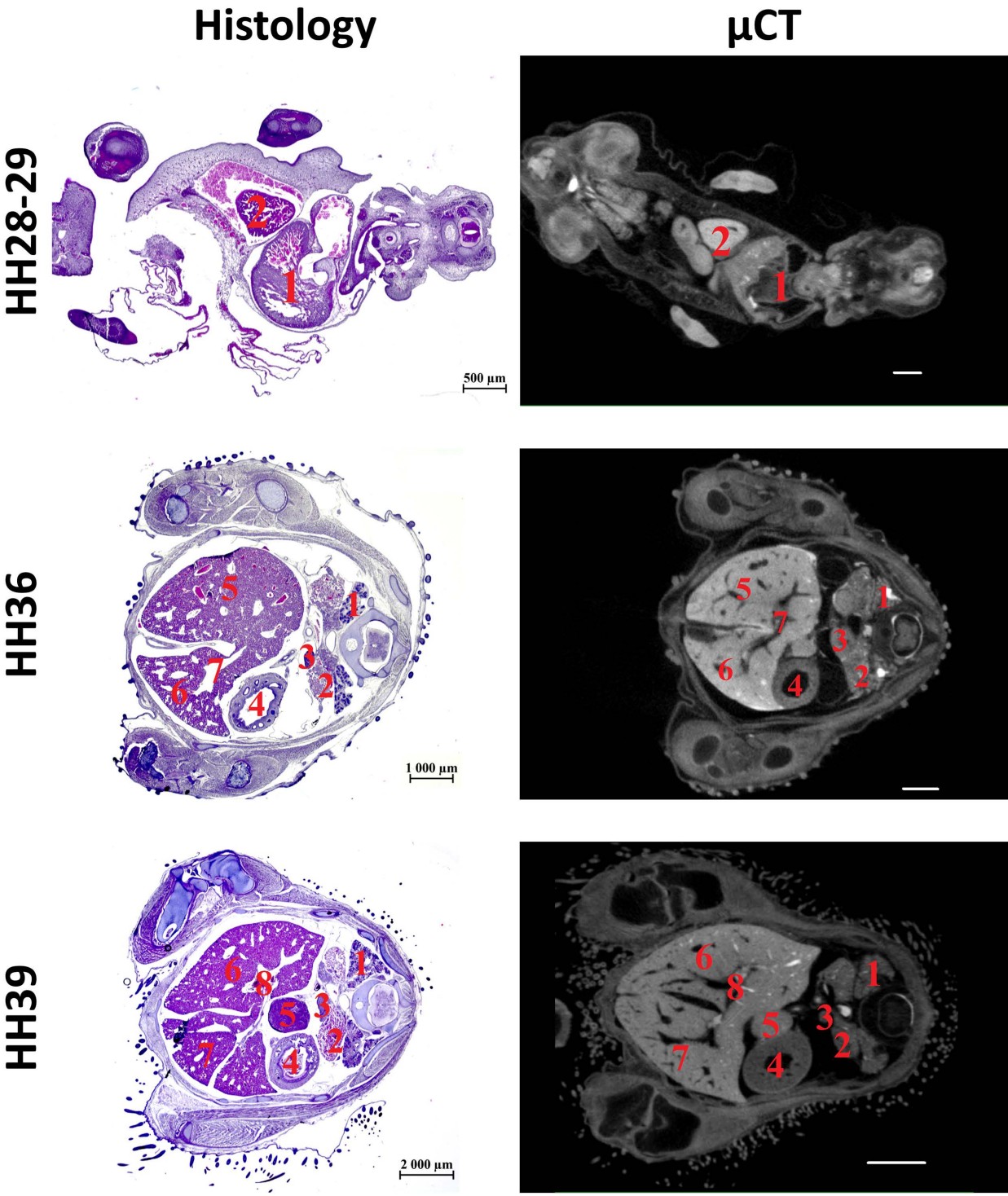

**Fig 2. Histological sections and representative sectional µCT images of the liver of a chick embryo at the 6th (HH28-HH29), 10th (HH36) and 13th (HH39) days of embryogenesis.** The following organs are marked on cross-sectional images of CE: day 6: heart (1), right lobe of the liver (2); day 10: metanephros (1), mesonephros (2), gonadal ridge (3), glandular stomach (4), right lobe of the liver (5), left lobe of the liver (6), portal vein (7); day 13: metanephrose (1), mesonephros (2), gonadal ridge (3), glandular stomach (4), spleen (5), right lobe of the liver (6), left lobe of the liver (7), portal vein (8). Scale ruler on µCT images: 500 µm (day 6), 1000 µm (day 10), 2000 µm (day 13).

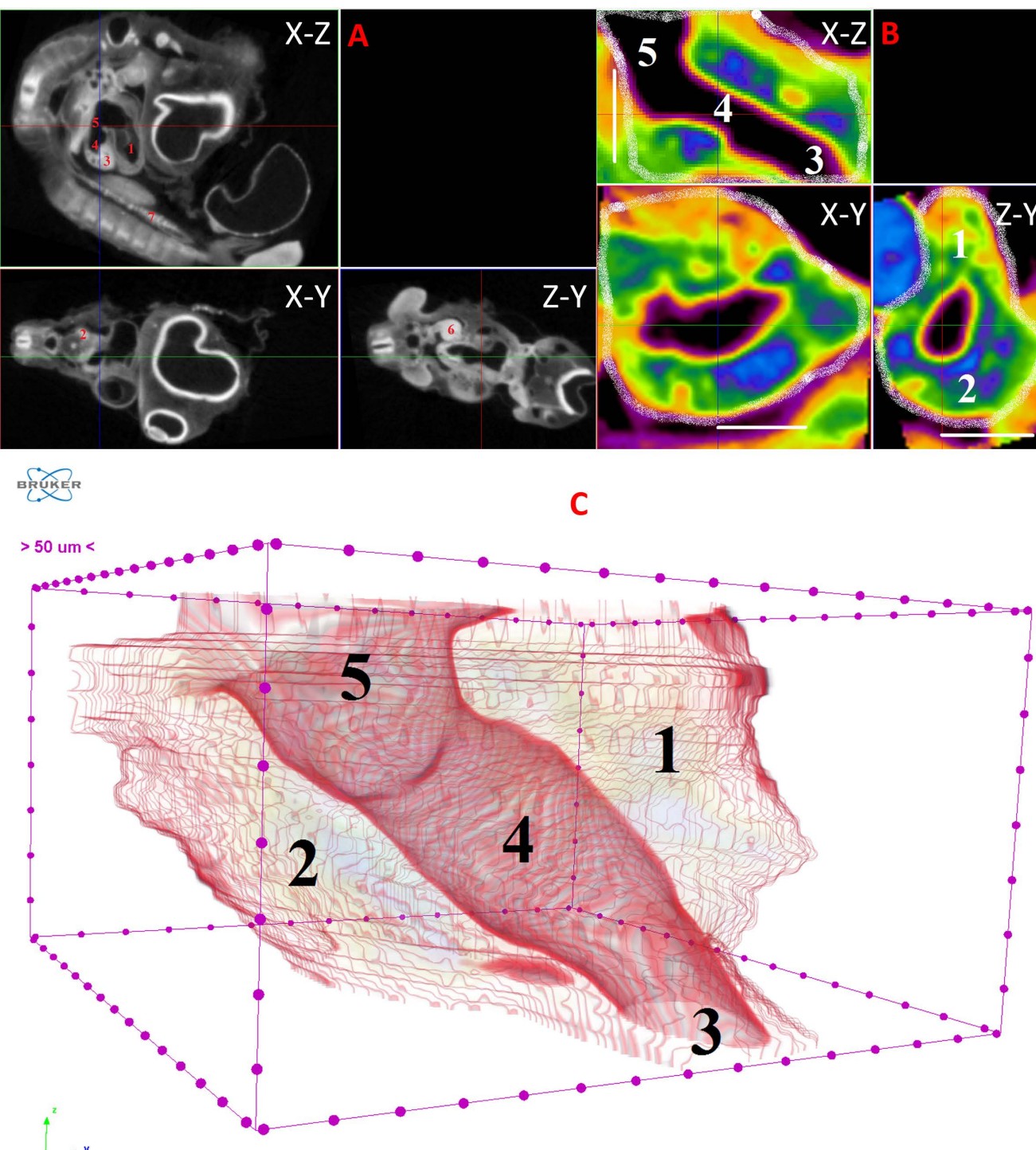

**Fig 3. Results of µCT of chick embryo on the 4th day (HH22-24) of embryogenesis.** (A) Representative cross-sectional images of the chick embryo: coronal (X-Z), sagittal (Z-Y), transaxial (X-Y) planes. The following organs are marked: heart (1), lungs (2), liver (3), ductus venosus (4), inferior vena cava (5), stomach (6), mesonephros (7). (B) Representative cross-sectional images of the liver: coronal (X-Z), transaxial (X-Y) and sagittal (Z-Y) planes. The following parts of the liver are marked: left lobe (1), right lobe (2), umbilical vein (3), ductus venosus (4), inferior vena cava (5). Scale ruler is 250 µm. The contours of the organ are highlighted with a white line in all three projections. (C) Isosurface 3D renderings of the liver and of the vessels liver. The following parts of the liver are marked: left lobe (1), right lobe (2), umbilical vein (3), ductus venosus (4), inferior vena cava (5).

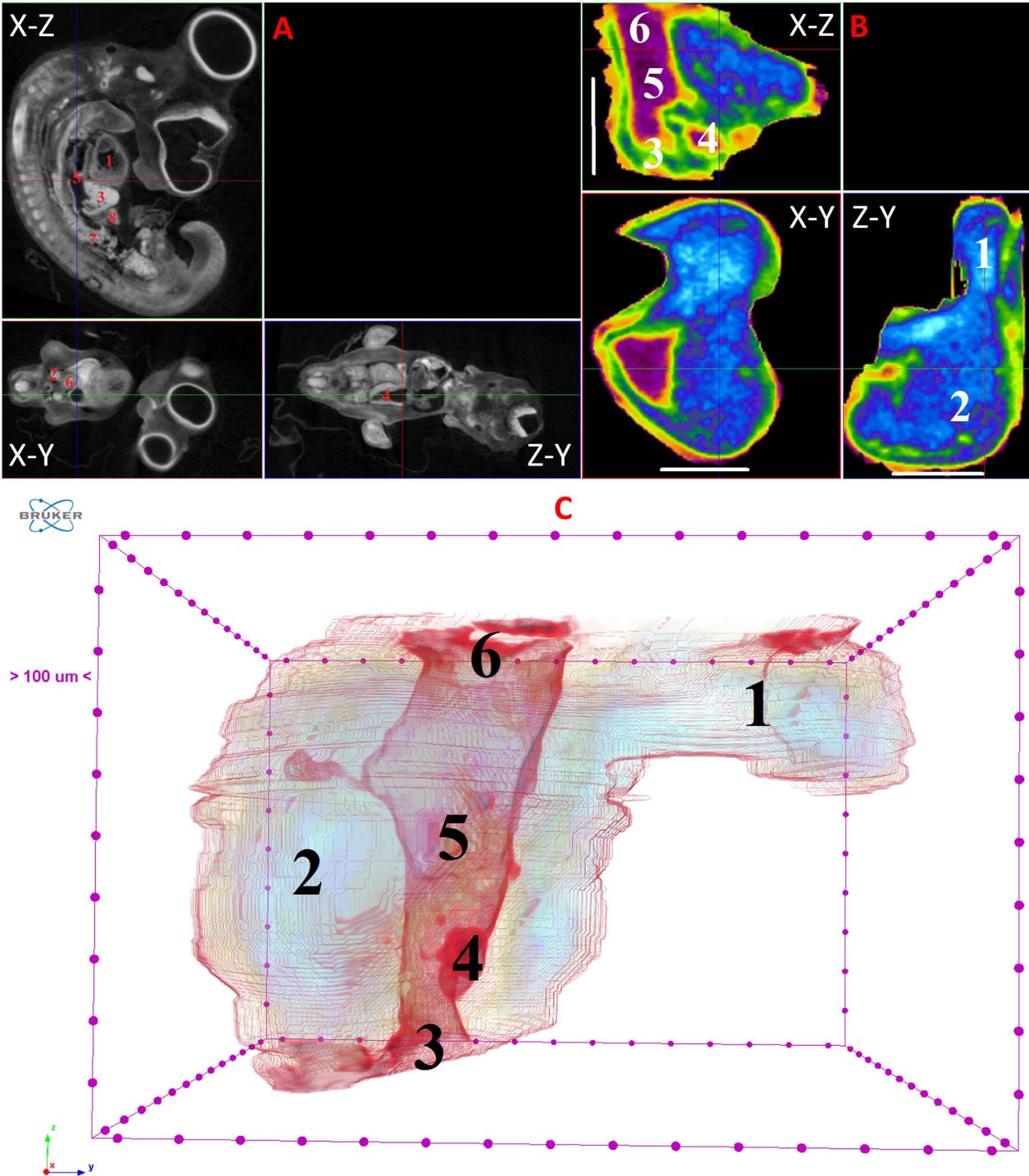

**Fig 4. Results of µCT of chick embryo on the 5th day (HH25-27) of embryogenesis.** (A) Representative cross-sectional images of the chick embryo: coronal (X-Z), sagittal (Z-Y), transaxial (X-Y) planes. The following organs are marked: heart (1), lungs (2), liver (3), ductus venosus (4), inferior vena cava (5), stomach (6), mesonephros (7), intestine (8). (B) Representative cross-sectional images of the liver: coronal (X-Z), transaxial (X-Y) and sagittal (Z-Y) planes. The following parts of the liver are marked: left lobe (1), right lobe (2), umbilical vein (3), portal vein (4), ductus venosus (5), inferior vena cava (6). Scale ruler is 500 µm. (C) Isosurface 3D renderings of the liver and of the vessels liver. The following parts of the liver are marked: left lobe (1), right lobe (2), umbilical vein (3), portal vein (4), ductus venosus (5), inferior vena cava (6).

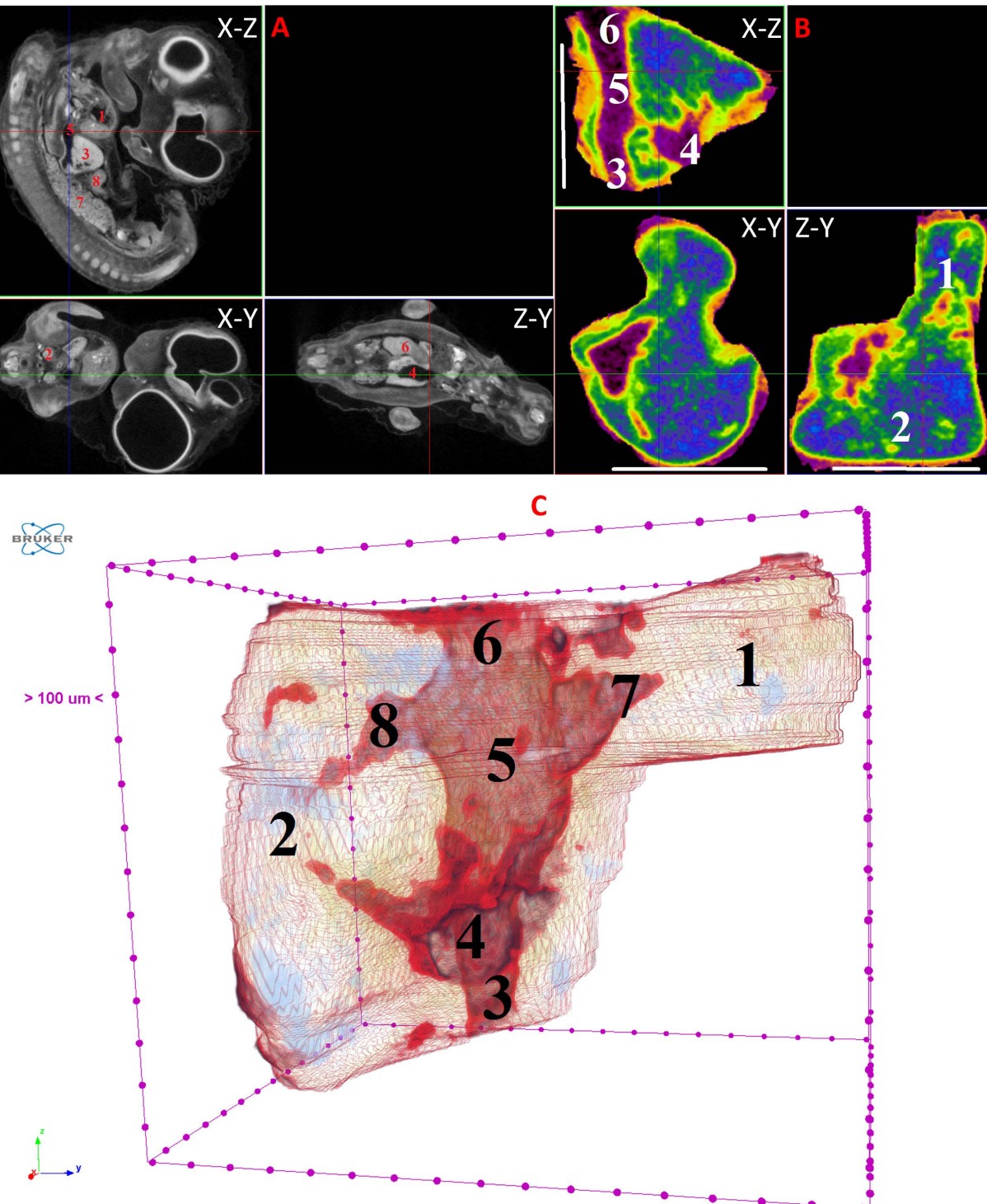

**Fig 5. Results of µCT of chick embryo on the 6th day (HH28-29) of embryogenesis.** (A) Representative cross-sectional images of the chick embryo: coronal (X-Z), sagittal (Z-Y), transaxial (X-Y) planes. The following organs are marked: heart (1), lungs (2), liver (3), ductus venosus (4), inferior vena cava (5), stomach (6), mesonephros (7), intestine (8). (B) Representative cross-sectional images of the liver (B): coronal (X-Z), transaxial (X-Y) and sagittal (Z-Y) planes. The following parts of the liver are marked: left lobe (1), right lobe (2), umbilical vein (3), portal vein (4), ductus venosus (5), inferior vena cava (6). Scale ruler is 1 mm. (C) Isosurface 3D renderings of the liver and of the vessels liver. The following parts of the liver are marked: left lobe (1), right lobe (2), umbilical vein (3), portal vein (4), ductus venosus (5), inferior vena cava (6), left hepatic vein (7), right hepatic vein (8).

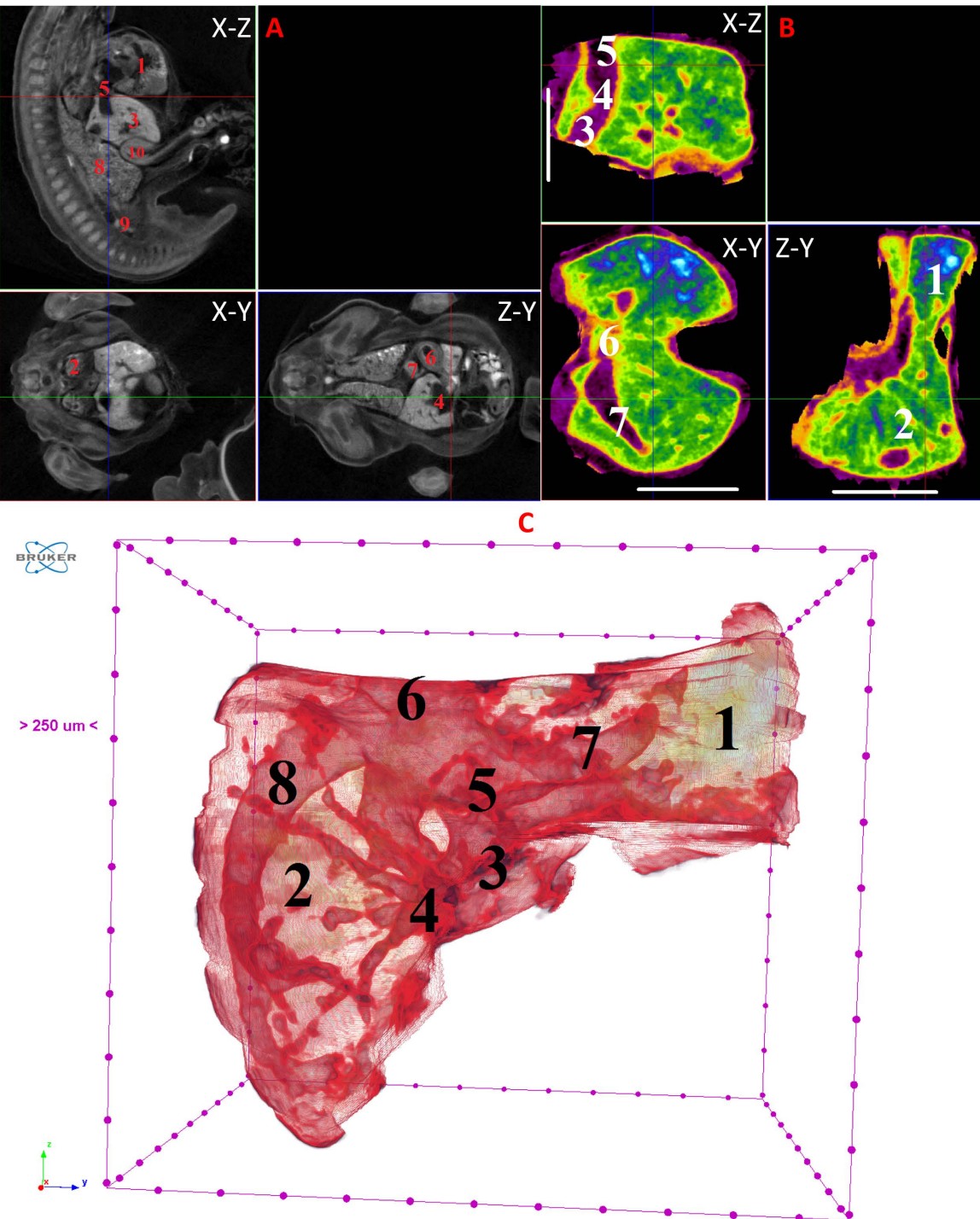

**Fig 6. Results of μCT of chick embryo on the 7th day (HH30-32) of embryogenesis.** (A) Representative cross-sectional images of the chick embryo: coronal (X-Z), sagittal (Z-Y), transaxial (X-Y) planes. The following organs are marked: heart (1), lungs (2), liver (3), ductus venosus (4), inferior vena cava (5), stomach (6), spleen (7), mesonephros (8), metanephros (9), intestine (10). (B) Representative cross-sectional images of the liver (B): coronal (X-Z), transaxial (X-Y) and sagittal (Z-Y) planes. The following parts of the liver are marked: left lobe (1), right lobe (2), umbilical vein (3), ductus venosus (4), inferior vena cava (5), left hepatic vein (6), right hepatic vein (7). Scale ruler is 1 mm. (C) Isosurface 3D renderings of the liver and of the vessels liver. The following parts of the liver are marked: left lobe (1), right lobe (2), umbilical vein (3), portal vein (4), ductus venosus (5), inferior vena cava (6), left hepatic vein (7), right hepatic vein (8).

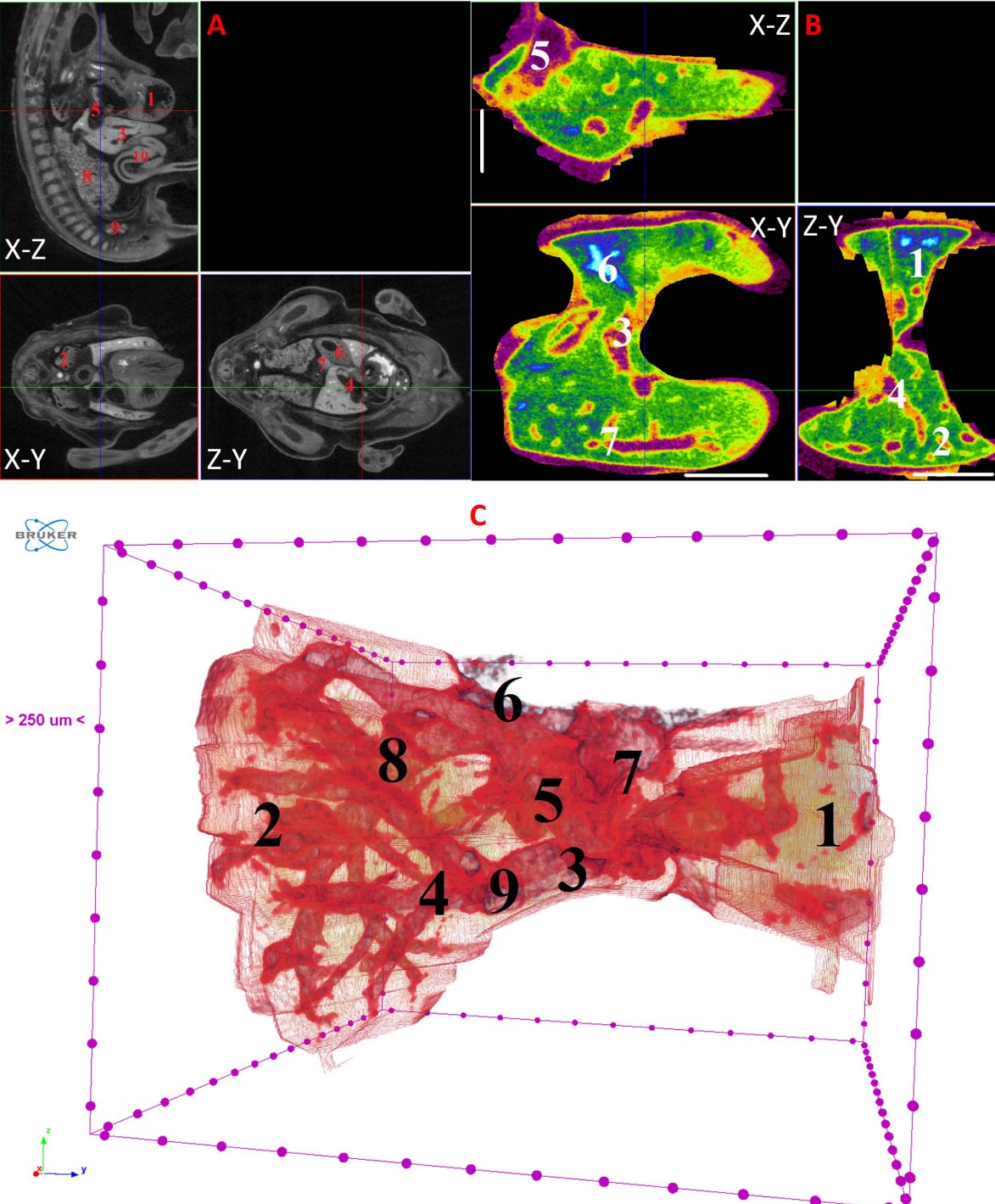

**Fig 7. Results of µCT of chick embryo on the 8th day (HH33-34) of embryogenesis.** (A) Representative cross-sectional images of the chick embryo: coronal (X-Z), sagittal (Z-Y), transaxial (X-Y) planes. The following organs are marked: heart (1), lungs (2), liver (3), ductus venosus (4), inferior vena cava (5), stomach (6), spleen (7), mesonephros (8), metanephros (9), intestine (10). (B) Representative cross-sectional images of the liver (B): coronal (X-Z), transaxial (X-Y) and sagittal (Z-Y) planes. The following parts of the liver are marked: left lobe (1), right lobe (2), umbilical vein (3), portal vein (branches) (4), inferior vena cava (5), left hepatic vein (6), right hepatic vein (7). Scale ruler is 1 mm. (C) Isosurface 3D renderings of the liver and of the vessels liver. The following parts of the liver are marked: left lobe (1), right lobe (2), umbilical vein (3), portal vein (branches) (4), ductus venosus (5), inferior vena cava (6). left hepatic vein (7), right hepatic vein (8), gallbladder (9).

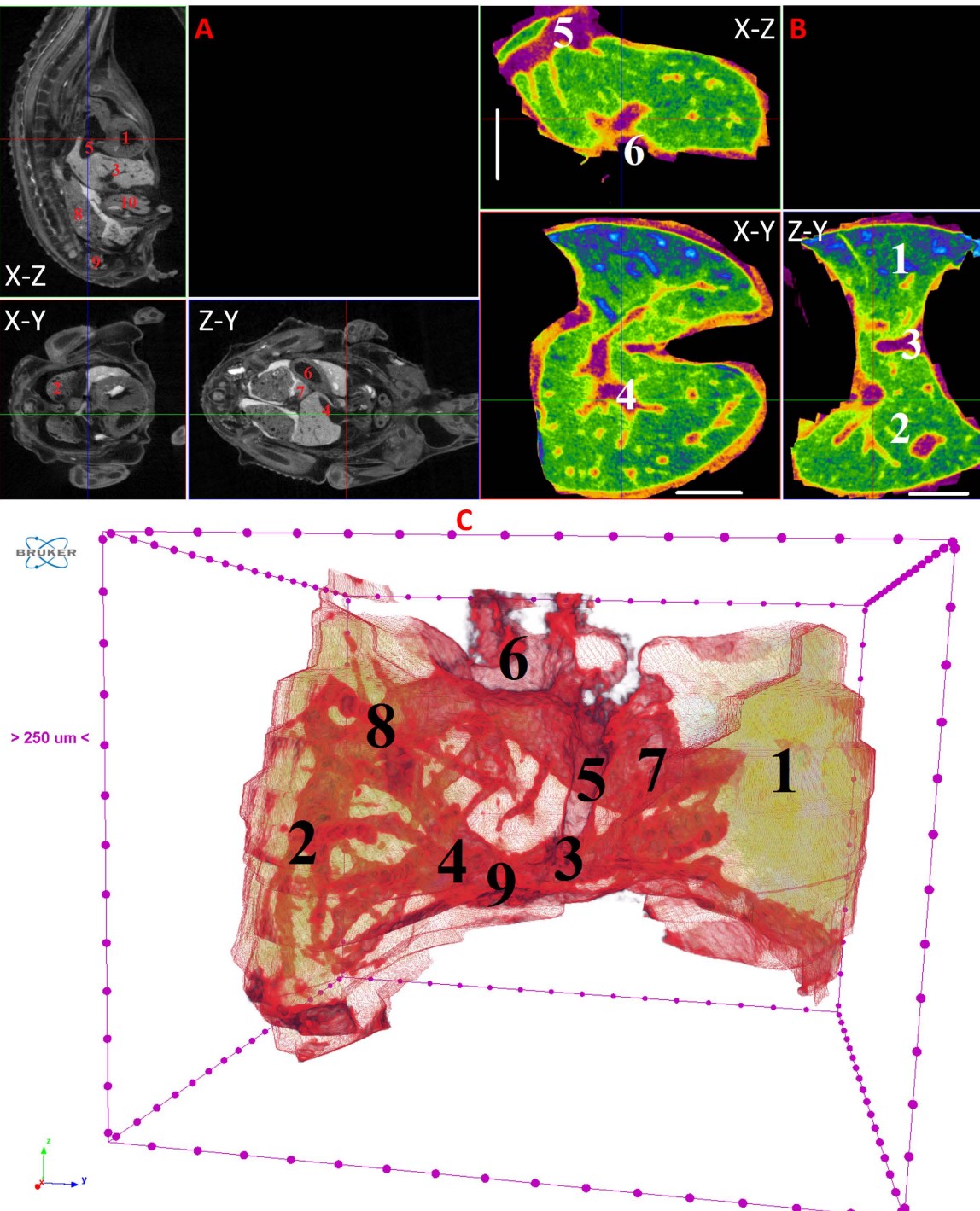

**Fig 8. Results of µCT of chick embryo on the 9th day (HH35) of embryogenesis.** (A) Representative cross-sectional images of the chick embryo: coronal (X-Z), sagittal (Z-Y), transaxial (X-Y) planes. The following organs are marked: heart (1), lungs (2), liver (3), ductus venosus (4), inferior vena cava (5), stomach (6), spleen (7), mesonephros (8), metanephros (9), intestine (10). (B) Representative cross-sectional images of the liver: coronal (X-Z), transaxial (X-Y) and sagittal (Z-Y) planes. The following parts of the liver are marked: left lobe (1), right lobe (2), umbilical vein (3), portal vein (4), inferior vena cava (5), gallbladder (6). Scale ruler is 1 mm. (C) Isosurface 3D renderings of the liver and of the vessels liver. The following parts of the liver are marked: left lobe (1), right lobe (2), umbilical vein (3), portal vein (4), ductus venosus (5), inferior vena cava (6), left hepatic vein (7), right hepatic vein (8), gallbladder (9).

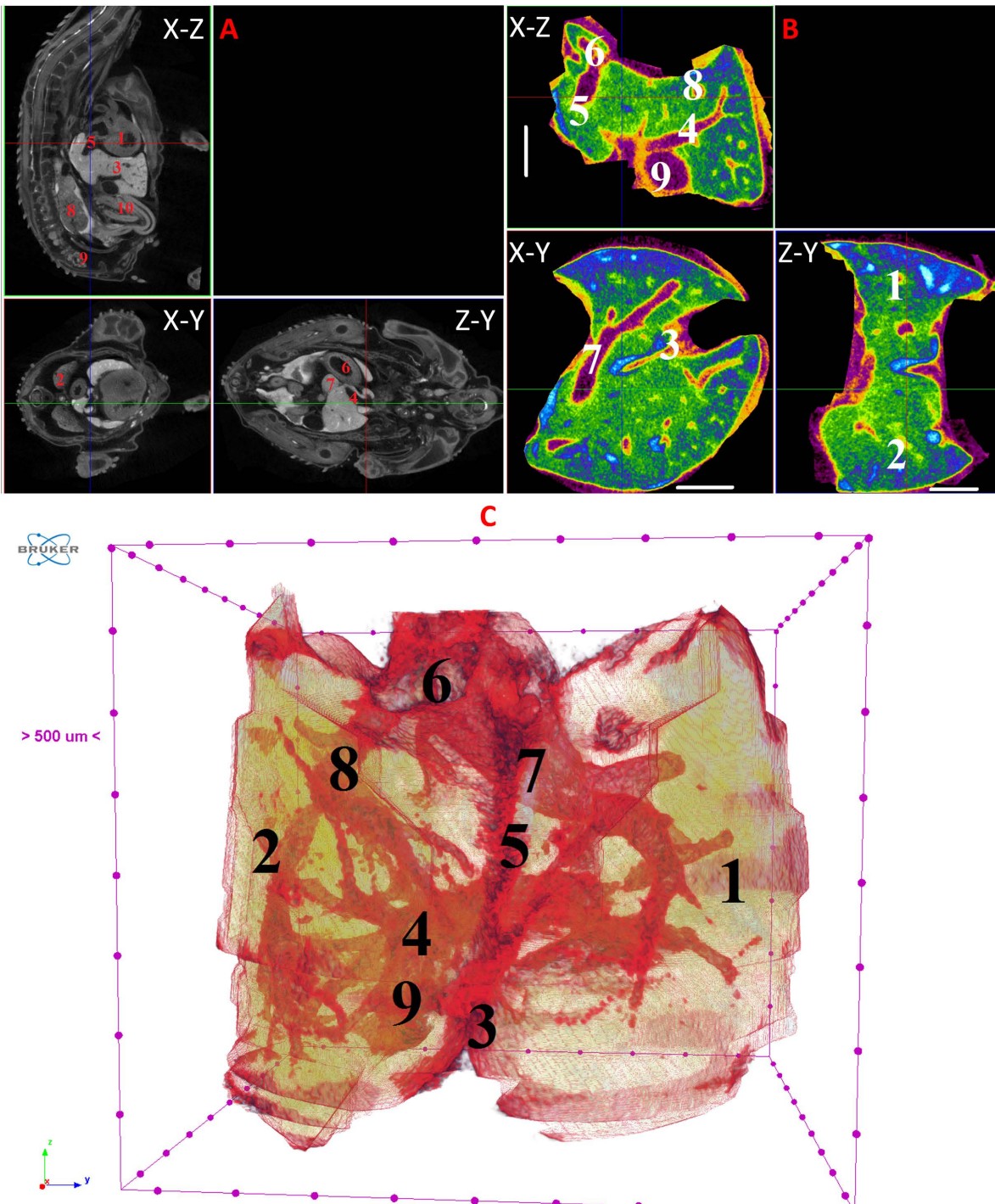

**Fig 9. Results of µCT of chick embryo on the 10th day (HH36) of embryogenesis.** (A) Representative cross-sectional images of the chick embryo: coronal (X-Z), sagittal (Z-Y), transaxial (X-Y) planes. The following organs are marked: heart (1), lungs (2), liver (3), ductus venosus (4), inferior vena cava (5), stomach (6), spleen (7), mesonephros (8), metanephros (9), intestine (10). (B) Representative cross-sectional images of the liver: coronal (X-Z), transaxial (X-Y) and sagittal (Z-Y) planes. The following parts of the liver are marked: left lobe (1), right lobe (2), umbilical vein (3), portal vein (4), ductus venosus (5), inferior vena cava (6), left hepatic vein (7), right hepatic vein (8), gallbladder (9). Scale ruler is 1 mm. (C) Isosurface 3D renderings of the liver and of the vessels liver. The following parts of the liver are marked: left lobe (1), right lobe (2), umbilical vein (3), portal vein (4), ductus venosus (5), inferior vena cava (6), left hepatic vein (7), right hepatic vein (8), gallbladder (9).

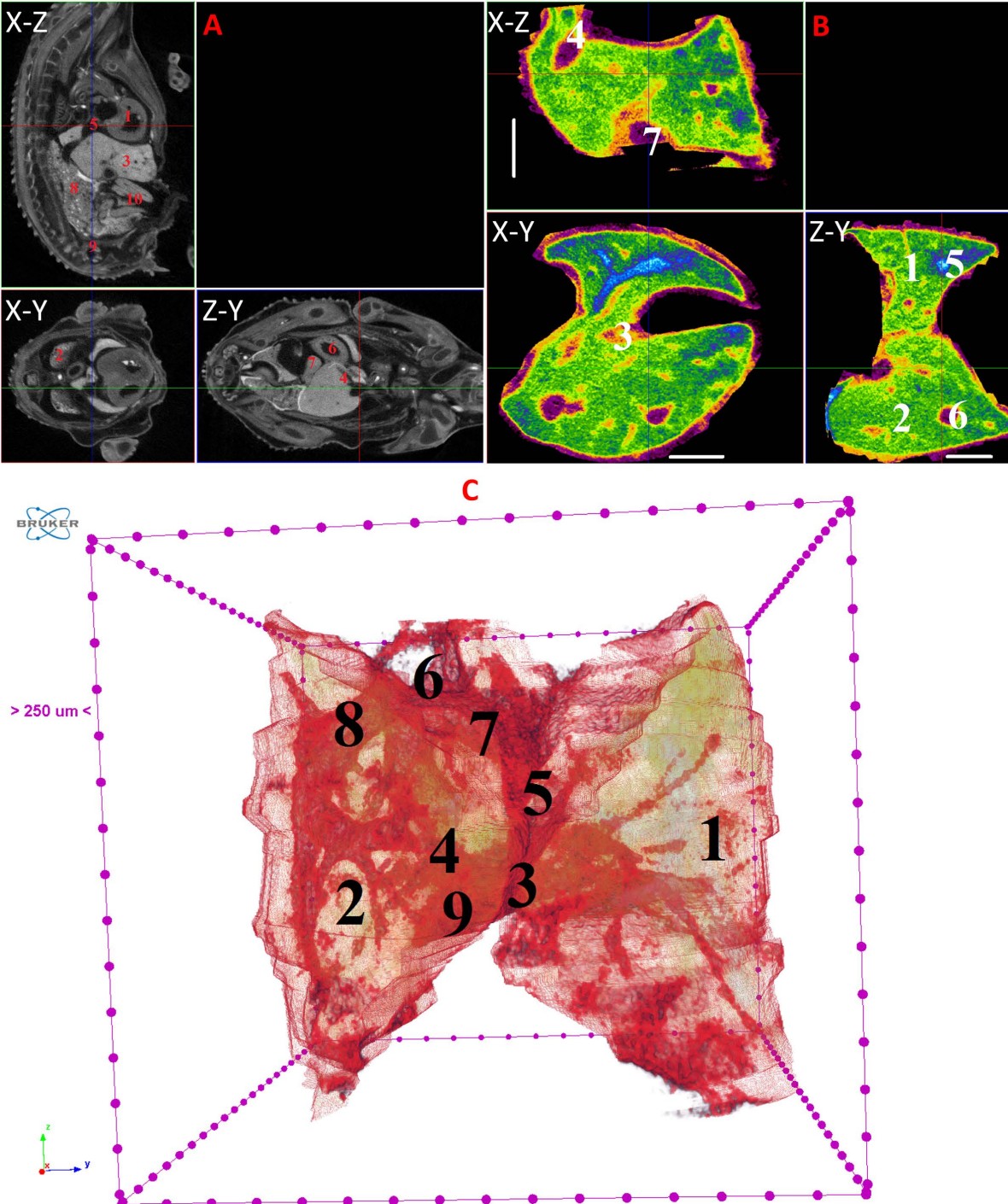

**Fig 10. Results of µCT of chick embryo on the 11ᵗʰ day (HH37) of embryogenesis.** (A) Representative cross-sectional images of the chick embryo: coronal (X-Z), sagittal (Z-Y), transaxial (X-Y) planes. The following organs are marked: heart (1), lungs (2), liver (3), ductus venosus (4), inferior vena cava (5), stomach (6), spleen (7), mesonephros (8), metanephros (9), intestine (10). (B) Representative cross-sectional images of the liver: coronal (X-Z), transaxial (X-Y) and sagittal (Z-Y) planes. The following parts of the liver are marked: left lobe (1), right lobe (2), ductus venosus (3), inferior vena cava (4), left hepatic vein (5), right hepatic vein (6), gallbladder (7). Scale ruler is 1 mm. (C) Isosurface 3D renderings of the liver and of the vessels liver. The following parts of the liver are marked: left lobe (1), right lobe (2), umbilical vein (3), portal vein (4), ductus venosus (5), inferior vena cava (6), left hepatic vein (7), right hepatic vein (8), gallbladder (9).

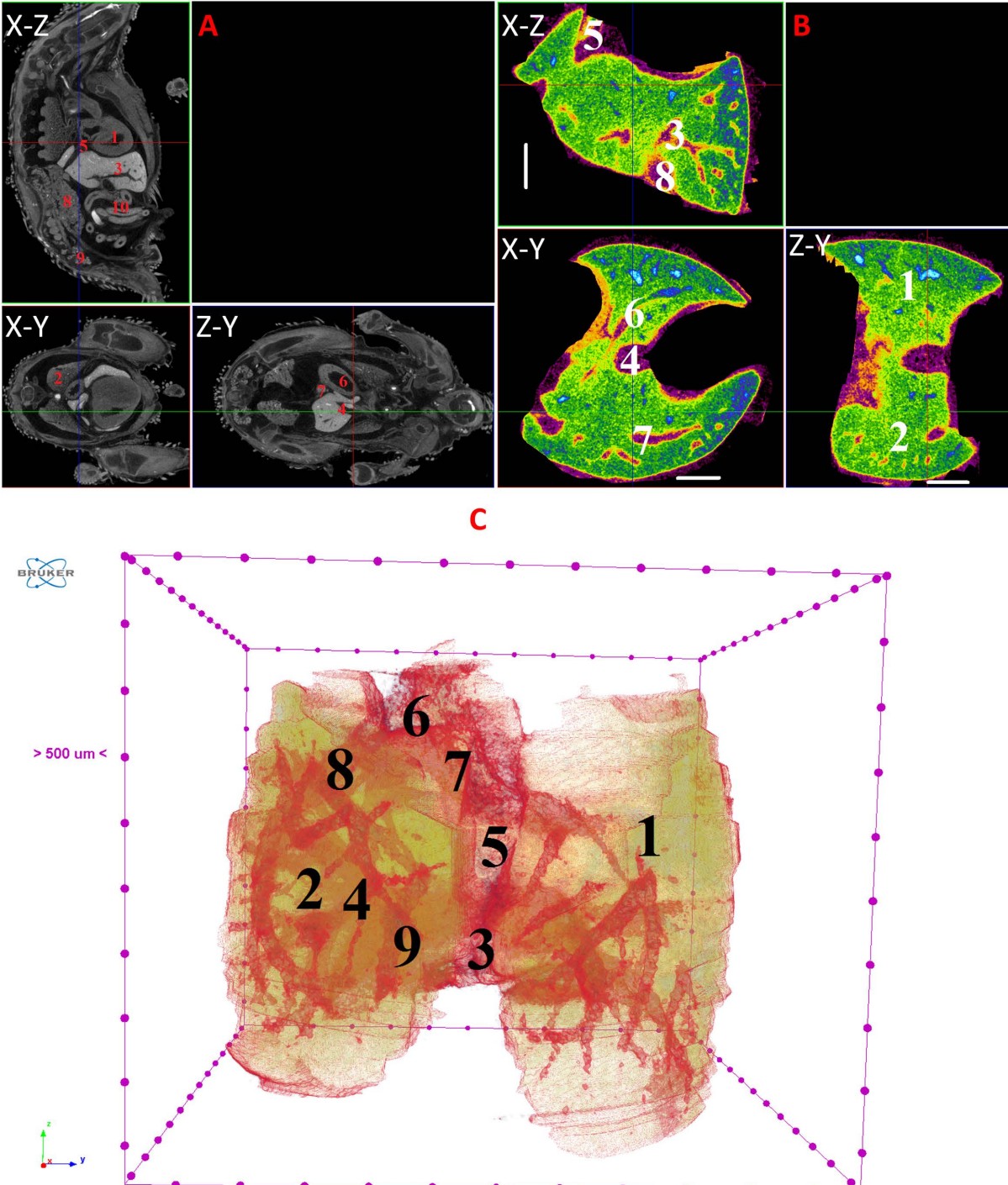

**Fig 11. Results of µCT of chick embryo on the 12th day (HH38) of embryogenesis.** (A) Representative cross-sectional images of the chick embryo: coronal (X-Z), sagittal (Z-Y), transaxial (X-Y) planes. The following organs are marked: heart (1), lungs (2), liver (3), ductus venosus (4), inferior vena cava (5), stomach (6), spleen (7), mesonephros (8), metanephros (9), intestine (10). (B) Representative cross-sectional images of the liver: coronal (X-Z), transaxial (X-Y) and sagittal (Z-Y) planes. The following parts of the liver are marked: left lobe (1), right lobe (2), portal vein (3), ductus venosus (4), inferior vena cava (5), left hepatic vein (6), right hepatic vein (7), gallbladder (8). Scale ruler is 1 mm. (C) Isosurface 3D renderings of the liver and of the vessels liver. The following parts of the liver are marked: left lobe (1), right lobe (2), umbilical vein (3), portal vein (4), ductus venosus (5), inferior vena cava (6), left hepatic vein (7), right hepatic vein (8), gallbladder (9).

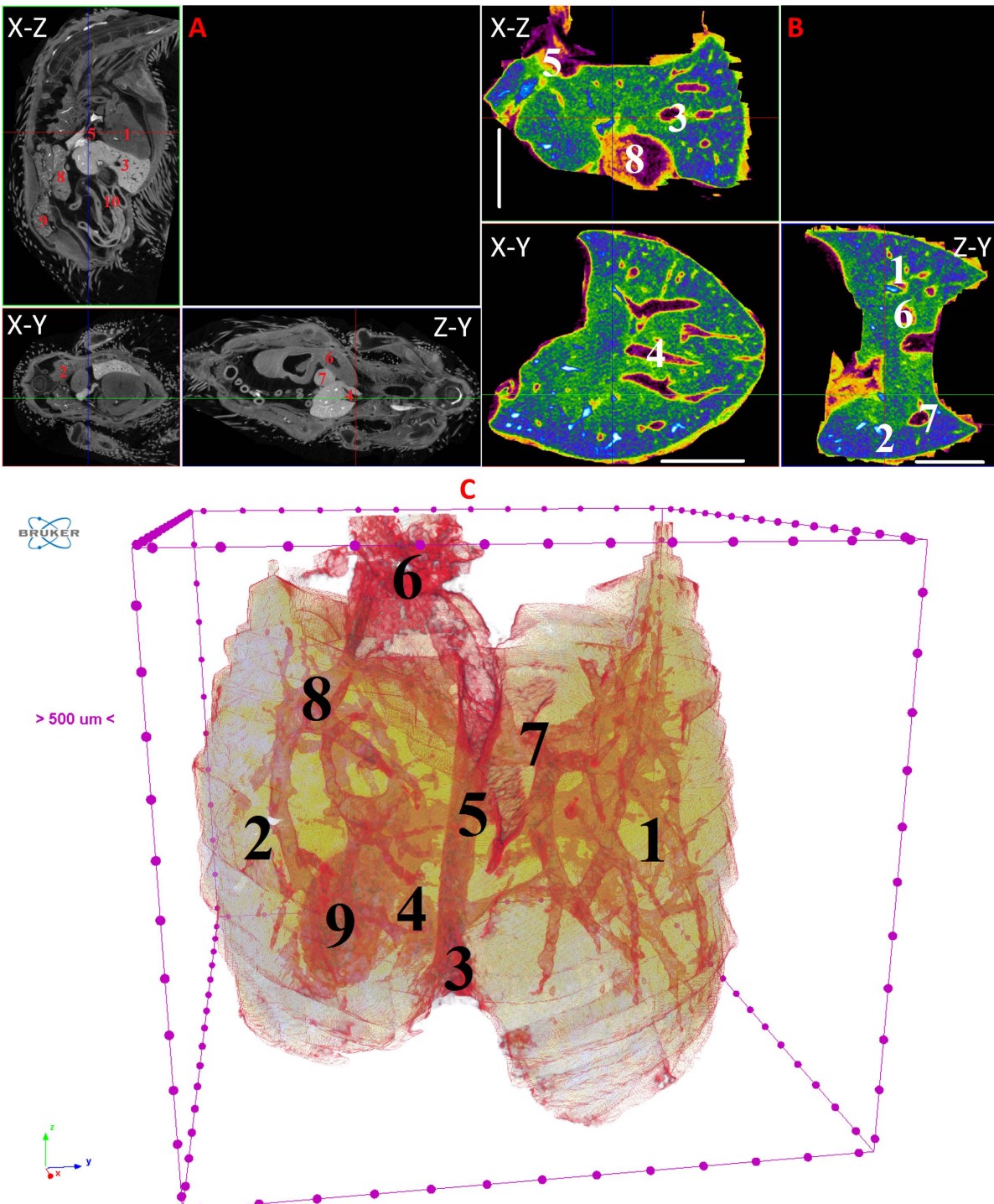

**Fig 12. Results of μCT of chick embryo on the 13th day (HH39) of embryogenesis.** (A) Representative cross-sectional images of the chick embryo: coronal (X-Z), sagittal (Z-Y), transaxial (X-Y) planes. The following organs are marked: heart (1), lungs (2), liver (3), ductus venosus (4), inferior vena cava (5), stomach (6), spleen (7), mesonephros (8), metanephros (9), intestine (10). (B) Representative cross-sectional images of the liver: coronal (X-Z), transaxial (X-Y) and sagittal (Z-Y) planes. The following parts of the liver are marked: left lobe (1), right lobe (2), portal vein (3), ductus venosus (4), inferior vena cava (5), left hepatic vein (6), right hepatic vein (7), gallbladder (8). Scale ruler is 2 mm. (C) Isosurface 3D renderings of the liver and of the vessels liver. The following parts of the liver are marked: left lobe (1), right lobe (2), umbilical vein (3), portal vein (4), ductus venosus (5), inferior vena cava (6), left hepatic vein (7), right hepatic vein (8), gallbladder (9).

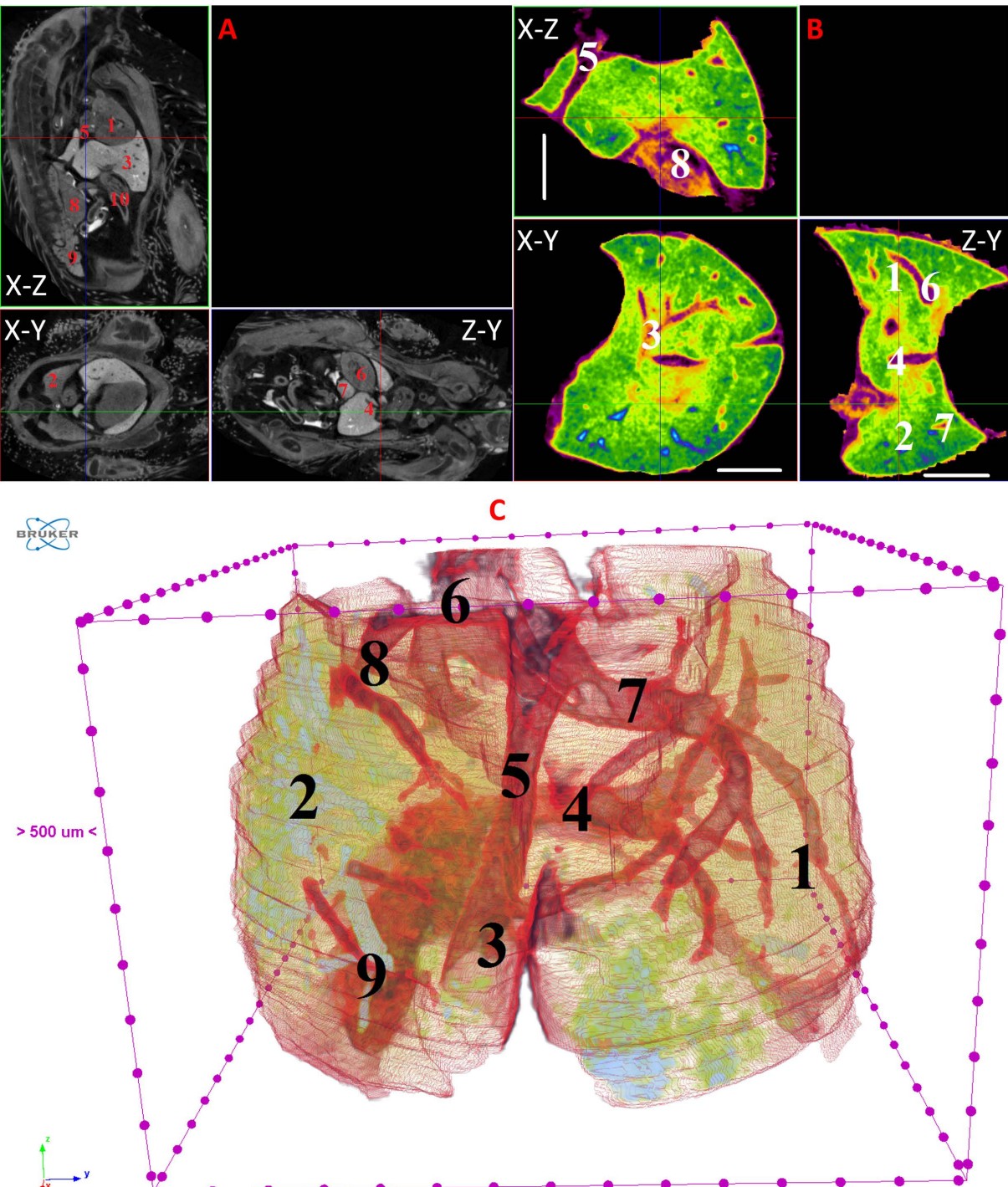

**Fig 13. Results of µCT of chick embryo on the 14th day (HH40) of embryogenesis.** (A) Representative cross-sectional images of the chick embryo: coronal (X-Z), sagittal (Z-Y), transaxial (X-Y) planes. The following organs are marked: heart (1), lungs (2), liver (3), ductus venosus (4), inferior vena cava (5), stomach (6), spleen (7), mesonephros (8), metanephros (9), intestine (10). (B) Representative cross-sectional images of the liver: coronal (X-Z), transaxial (X-Y) and sagittal (Z-Y) planes. The following parts of the liver are marked: left lobe (1), right lobe (2), portal vein (3), ductus venosus (4), inferior vena cava (5), left hepatic vein (6), right hepatic vein (7), gallbladder (8). Scale ruler is 2 mm. (C) Isosurface 3D renderings of the liver and of the vessels liver. The following parts of the liver are marked: left lobe (1), right lobe (2), umbilical vein (3), portal vein (4), ductus venosus (5), inferior vena cava (6), left hepatic vein (7), right hepatic vein (8), gallbladder (9).

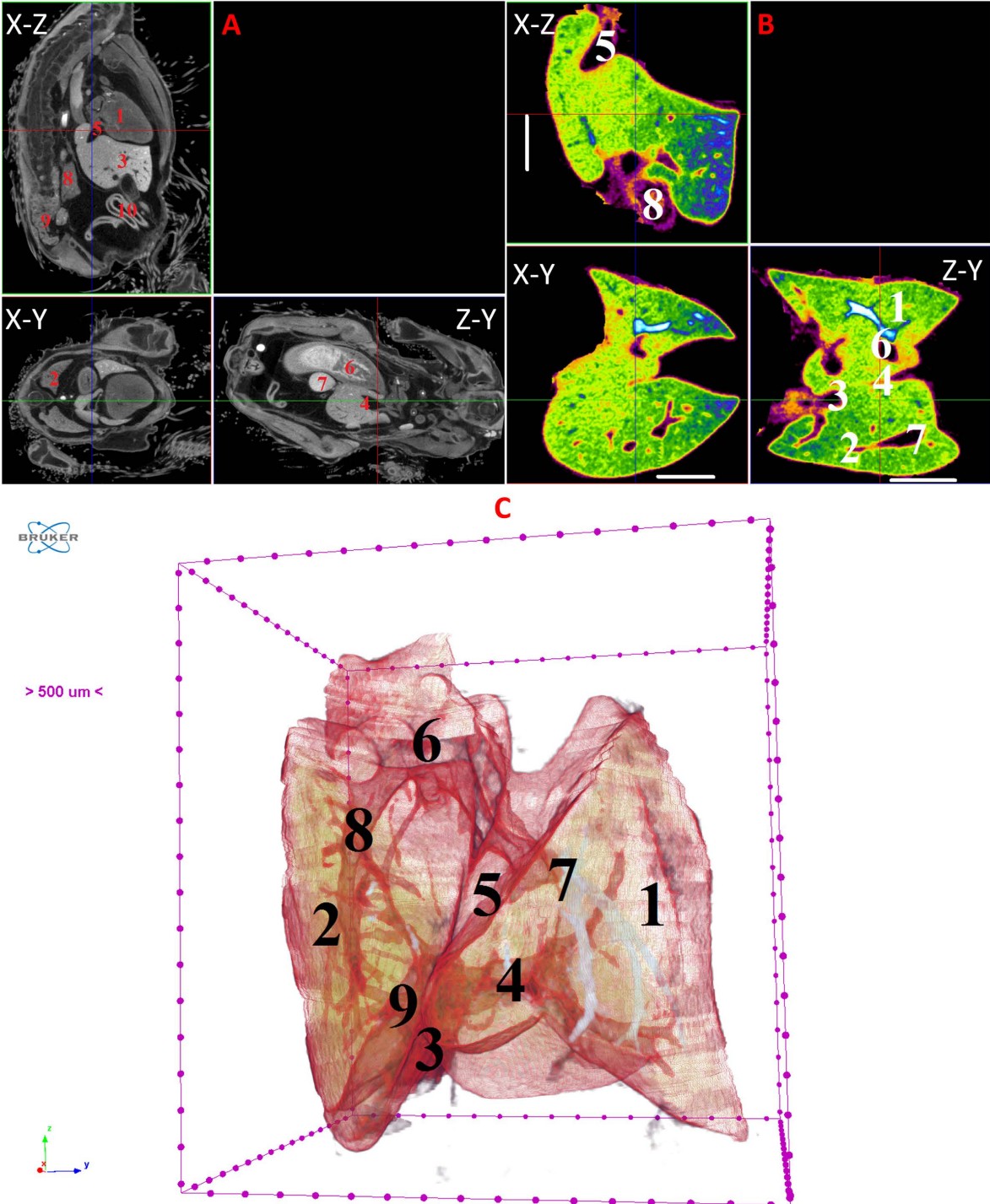

**Fig 14. Results of μCT of chick embryo on the 15th day (HH41) of embryogenesis.** (A) Representative cross-sectional images of the chick embryo: coronal (X-Z), sagittal (Z-Y), transaxial (X-Y) planes. The following organs are marked on cross-sectional images of CE: heart (1), lungs (2), liver (3), ductus venosus (4), inferior vena cava (5), stomach (6), spleen (7), mesonephros (8), metanephros (9), intestine (10). (B) Representative cross-sectional images of the liver: coronal (X-Z), transaxial (X-Y) and sagittal (Z-Y) planes. The following parts of the liver are marked: left lobe (1), right lobe (2), umbilical vein (3), ductus venosus (4), inferior vena cava (5), left hepatic vein (6), right hepatic vein (7), gallbladder (8). Scale ruler is 2 mm. (C) Isosurface 3D renderings of the liver and of the vessels liver. The following parts of the liver are marked: left lobe (1), right lobe (2), umbilical vein (3), portal vein (4), ductus venosus (5), inferior vena cava (6), left hepatic vein (7), right hepatic vein (8), gallbladder (9).

occupies a substantial volume. Other smaller vessels were not visualized due to the limitations of the µCT resolution (9 µm). Notably, on the 4th day, the liver is located in close proximity to the heart and stomach, with minimal interstitial space between these organs (Fig 3).

During the 5th and 6th days of incubation (HH25-HH29), the liver undergoes a significant increase in size, attributed to the growth of both parenchyma and stroma, including blood vessels. The right lobe of the liver exhibits substantial growth, and the initial segments of hepatic veins begin to appear. The spatial relationships between the liver and its surrounding organs become more clearly defined (Figs 4 and 5).

By the 7th day of incubation (HH30-HH32), pronounced growth and development of blood vessels are evident within the liver, with the total liver volume increasing significantly more rapidly than the total volume of the CE. The volume of the right lobe of the liver continues to expand intensively. In the 3D isosurface format, the left lobe, right lobe, umbilical vein, portal vein, ductus venosus, inferior vena cava, left hepatic vein, and right hepatic vein are readily visualized (Fig 6).

On the 8th and 9th days of incubation (HH33-HH35), the branching of smaller vessels, particularly the branches of the portal vein, begins to be visualized within the liver. A small gallbladder also becomes discernible. The left lobe of the liver starts to increase in size (Figs 7 and 8).

By day 10 (HH36) and continuing up to day 15 (HH37–41), the liver attains a more classic morphological configuration, with a significant increase in the size of the left lobe and the gallbladder. The regions of the hepatic veins and the branching pattern of the portal vein are well-delineated and visualized (Figs 9–14). These findings collectively present significant opportunities and can serve as a foundation for future fundamental and applied investigations into liver development during embryogenesis

## 4. Conclusions

This study provided a comprehensive 2D and 3D µCT analysis of chicken embryo liver development (HH22–HH41), enabled by a refined 1% PTA staining protocol that yielded unprecedented detail in microstructures, notably the vasculature. This demonstrated a qualitative congruence with histological analysis, hence our findings established that µCT is a robust method for high-resolution, imaging of the developing liver. We precisely mapped the dynamics of liver growth and identified a critical period of rapid expansion between days 6 and 9 of incubation, alongside the subsequent stabilization of vascular volume by day 10. Furthermore, the ease of structural orientation afforded by µCT sections and 3D renderings underscored its utility over traditional histological approach. By this we established a methodology for obtaining normative morphometric data, including the liver's spatial relationship with surrounding organs, its evolving morphology, and the intricate patterns of its vasculature. This research has unlocked significant new avenues for investigation in modern embryology, teratology, pharmacology, and toxicology.

### 4.1 Limitations of the study

In this work, we conducted a µCT analysis of the liver of the CE at HH22-HH41 embryonic stages. We considered it inappropriate to study the early stages of embryogenesis using this technique due to the insufficient detail of microtomograms. µCT analysis of the CE at HH42-HH46 embryonic stages also requires improved approaches to contrast, which is associated with the large size of the object.

### Acknowledgments

The study was conducted within the framework of the strategic academic leadership program "Priority-2030" using the equipment of the Center for Collective Use of the North Caucasus Federal University

### Author contributions

**Conceptualization:** Igor Rzhepakovsky.

**Data curation:** Igor Rzhepakovsky.

**Formal analysis:** Igor Rzhepakovsky, Sergei Piskov, Gloria Nassali, Idrisa Kiryowa, Andrey Nagdalian.

**Funding acquisition:** Igor Rzhepakovsky.

**Investigation:** Igor Rzhepakovsky, Sergei Piskov, Svetlana Avanesyan, Marina Sizonenko, Magomed Shakhbanov, Idrisa Kiryowa.

**Methodology:** Igor Rzhepakovsky, Sergei Piskov, Svetlana Avanesyan, Marina Sizonenko, Lyudmila Timchenko.

**Project administration:** Andrey Nagdalian.

**Resources:** Lyudmila Timchenko.

**Software:** Igor Rzhepakovsky.

**Supervision:** Lyudmila Timchenko.

**Validation:** Gloria Nassali, Andrey Nagdalian.

**Visualization:** Igor Rzhepakovsky, Sergei Piskov.

**Writing – original draft:** Igor Rzhepakovsky, Sergei Piskov, Svetlana Avanesyan, Marina Sizonenko, Magomed Shakhbanov, Gloria Nassali.

**Writing – review & editing:** Lyudmila Timchenko, Gloria Nassali, Idrisa Kiryowa, Andrey Nagdalian.

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
