## [Decision Letter · Decision Letter 0]

7 Jul 2025

Dear Dr. Nassali,

Thank you for submitting your manuscript to PLOS ONE. After careful consideration, we feel that it has merit but does not fully meet PLOS ONE’s publication criteria as it currently stands. Therefore, we invite you to submit a revised version of the manuscript that addresses the points raised during the review process.

We look forward to receiving your revised manuscript.

Kind regards,

Ayman A Swelum

Academic Editor

PLOS ONE

Journal Requirements:

4. In the online submission form you indicate that your data is not available for proprietary reasons and have provided a contact point for accessing this data. Please note that your current contact point is a co-author on this manuscript. According to our Data Policy, the contact point must not be an author on the manuscript and must be an institutional contact, ideally not an individual. Please revise your data statement to a non-author institutional point of contact, such as a data access or ethics committee, and send this to us via return email. Please also include contact information for the third party organization, and please include the full citation of where the data can be found.

6. Please include a copy of Table 1 which you refer to in your text on page 17.

Additional Editor Comments:

** Please respond carefully for all reviewers comments.**

Reviewers' comments:

Reviewer's Responses to Questions

**Comments to the Author**

1. Is the manuscript technically sound, and do the data support the conclusions?

Reviewer #1: Yes

Reviewer #2: Yes

Reviewer #3: Partly

2. Has the statistical analysis been performed appropriately and rigorously?

Reviewer #1: I Don't Know

Reviewer #2: Yes

Reviewer #3: Yes

3. Have the authors made all data underlying the findings in their manuscript fully available?

Reviewer #1: Yes

Reviewer #2: Yes

Reviewer #3: No

4. Is the manuscript presented in an intelligible fashion and written in standard English?

Reviewer #1: Yes

Reviewer #2: Yes

Reviewer #3: Yes

Reviewer #1: The manuscript is scientifically sound, methodologically rigorous, and clearly presented.

Minor revisions are needed to enhance clarity and reproductibility.The imaging methodology is of broad potential interest for developmental biology and non-invasive phenotyping, especially in model organisms.

Reviewer #2: Very well executed and designed radiological study of liver morphogenesis. Figures comparing histological slides and uCT scans are really impressive. I recommend the manuscript for publishing since it is novel in the field of embryology.

Reviewer #3: In their paper, 'Rapid 3D phenotyping of chick embryo liver development at HH22-HH41 embryonic stages using X-ray microcomputed tomography with PTA staining', the authors discuss the use of µCT analysis as a possible tool for monitoring changes in liver tissue during the development of the chick embryo as an animal model. The authors employed 1% phosphotungstic acid (PTA) as a contrast agent. This method has been used in other studies, but on different organs or animal models, or with a different concentration or duration of staining. The staining method was described in sufficient detail to allow it to be repeated. µCT images were visually compared directly with stained histological sections of embryos at corresponding developmental stages. By analysing the µCT images, the authors obtained data on liver and vascular volume at different stages of development. These data should have been summarised in Table 1, which the authors mention in the text but do not attach. The work is well written and the cited sources are up to date and relevant to the topic. The text is written in an intelligible fashion and in standard English.

Major issues:

Add Table 1, which contains data on liver and liver vessel volumes at various stages of chicken development.

Minor issues:

1. I propose adding the units in which the liver volume and vessel volume values are expressed to the caption of Figure 1.

2. I also propose replacing the term 'kidney' with 'metanephros' in the captions of the figures (Fig. 2, 6, 7, 8, 9, 10, 11, 12, 13 and 14), as used in Doaa et al. (2013) for similar developmental stages of chick embryos, for example. The term 'metanephros' more accurately describes the incomplete development of the kidney.

Doaa, M. Mokhtar; Enas, A. El-Hafez; Hassan, A. H.S; Fatma, A, Mostafa. Dynamics of Liver Development in Dandarawi Chicken. Journal of World's Poultry Research, Volume 3, Number 3, 2013, pp. 73-79(7)

3. Finally, I suggest omitting the “quantitative” congruence with the histological and uCT analyses described in the 'Conclusions' section. The manuscript does not describe quantitative histological analysis, e.g. using stereology or image analysis of histological slides.

**Do you want your identity to be public for this peer review?** For information about this choice, including consent withdrawal, please see our Privacy Policy

Reviewer #1: No

Reviewer #2: **Yes: ** Benjamin Benzon

Reviewer #3: No

---

## [Author Response · Author response to Decision Letter 1]

6 Sep 2025

We appreciate the Editor and Reviewers for the time devoted for study of our manuscript and valuable comments and recommendations that helped us to improve the quality of our work. All comments were considered and decided correspondingly. All revised parts are marked by color. Please find point-by-point responses below.

Reviewer #1: The manuscript is scientifically sound, methodologically rigorous, and clearly presented.

Minor revisions are needed to enhance clarity and reproductibility.The imaging methodology is of broad potential interest for developmental biology and non-invasive phenotyping, especially in model organisms.

Response: Thank you very much for the positive response on our study. Regarding clarity and reproductibility we carefully checked the Materials and methods section for essential information on equipment, software and chemicals, as well as statistical data processing.

Reviewer #2: Very well executed and designed radiological study of liver morphogenesis. Figures comparing histological slides and uCT scans are really impressive. I recommend the manuscript for publishing since it is novel in the field of embryology.

Response: Thank you very much for the positive response on our study. We value it!

Reviewer #3: In their paper, 'Rapid 3D phenotyping of chick embryo liver development at HH22-HH41 embryonic stages using X-ray microcomputed tomography with PTA staining', the authors discuss the use of µCT analysis as a possible tool for monitoring changes in liver tissue during the development of the chick embryo as an animal model. The authors employed 1% phosphotungstic acid (PTA) as a contrast agent. This method has been used in other studies, but on different organs or animal models, or with a different concentration or duration of staining. The staining method was described in sufficient detail to allow it to be repeated. µCT images were visually compared directly with stained histological sections of embryos at corresponding developmental stages. By analysing the µCT images, the authors obtained data on liver and vascular volume at different stages of development. These data should have been summarised in Table 1, which the authors mention in the text but do not attach. The work is well written and the cited sources are up to date and relevant to the topic. The text is written in an intelligible fashion and in standard English.

Major issues:

Add Table 1, which contains data on liver and liver vessel volumes at various stages of chicken development.

Response: Thank you for your attentiveness and sorry for missing Table 1. Now it was attached

Minor issues:

1. I propose adding the units in which the liver volume and vessel volume values are expressed to the caption of Figure 1.

Response: Thank you for recommendation. Figure 1 was revised

2. I also propose replacing the term 'kidney' with 'metanephros' in the captions of the figures (Fig. 2, 6, 7, 8, 9, 10, 11, 12, 13 and 14), as used in Doaa et al. (2013) for similar developmental stages of chick embryos, for example. The term 'metanephros' more accurately describes the incomplete development of the kidney.

Doaa, M. Mokhtar; Enas, A. El-Hafez; Hassan, A. H.S; Fatma, A, Mostafa. Dynamics of Liver Development in Dandarawi Chicken. Journal of World's Poultry Research, Volume 3, Number 3, 2013, pp. 73-79(7)

Response: We accept the remark with gratitude. It is a good article, we referred to it in the materials and methods as a source on the basis of which we labeled and described the organs in 2.5. Histological preparation. It is clearly seen there that it is better to write metanephros, since the kidney at the stages of embryogenesis under consideration is still underdeveloped. We agree!

3. Finally, I suggest omitting the “quantitative” congruence with the histological and uCT analyses described in the 'Conclusions' section. The manuscript does not describe quantitative histological analysis, e.g. using stereology or image analysis of histological slides.

Response: Thank you for recommendation. We removed “qualitative” word

---

## [Decision Letter · Decision Letter 1]

22 Sep 2025

Rapid 3D phenotyping of chick embryo liver development at HH22-HH41 embryonic stages using X-ray microcomputed tomography with PTA staining

PONE-D-25-19847R1

Dear Dr. Nassali,

We’re pleased to inform you that your manuscript has been judged scientifically suitable for publication and will be formally accepted for publication once it meets all outstanding technical requirements.

Kind regards,

Ayman A Swelum

Academic Editor

PLOS ONE

Additional Editor Comments (optional):

Reviewer #1:

Reviewer #2:

Reviewer #3:

Reviewers' comments:

Reviewer's Responses to Questions

**Comments to the Author**

Reviewer #1: All comments have been addressed

Reviewer #2: All comments have been addressed

Reviewer #3: All comments have been addressed

2. Is the manuscript technically sound, and do the data support the conclusions?

Reviewer #1: Yes

Reviewer #2: Yes

Reviewer #3: Yes

3. Has the statistical analysis been performed appropriately and rigorously?

Reviewer #1: I Don't Know

Reviewer #2: Yes

Reviewer #3: Yes

4. Have the authors made all data underlying the findings in their manuscript fully available?

Reviewer #1: Yes

Reviewer #2: Yes

Reviewer #3: Yes

5. Is the manuscript presented in an intelligible fashion and written in standard English?

Reviewer #1: Yes

Reviewer #2: Yes

Reviewer #3: Yes

Reviewer #1: (No Response)

Reviewer #2: All of the concerns raised by fellow reviewer were successfully addressed, so in my opinion the manuscript should be accepted for publication.

Reviewer #3: I would like to thank the authors for incorporating my suggestions. The study is very well executed and designed, presenting a high-quality radiological examination of liver morphogenesis, comparable to histological slides. The imaging methodology has broad potential significance for developmental biology and non-invasive phenotyping, particularly in model organisms. I recommend the manuscript for publication.

**Do you want your identity to be public for this peer review?** For information about this choice, including consent withdrawal, please see our Privacy Policy

Reviewer #1: No

Reviewer #2: **Yes: ** Benjamin Benzon

Reviewer #3: No

---

## [Editor Report · Acceptance letter]

PONE-D-25-19847R1

PLOS ONE

Dear Dr. Nassali,

I'm pleased to inform you that your manuscript has been deemed suitable for publication in PLOS ONE. Congratulations! Your manuscript is now being handed over to our production team.

Kind regards,

on behalf of

Professor Ayman A Swelum

Academic Editor

PLOS ONE